# The impact of mixed-ownership reform on zombie firms: Evidence from Chinese listed SOEs

Yufei Yin[1], Kexin Cao[2]*

1 School of Economics and Management, Huzhou University, Huzhou, China, 2 Faculty of Arts and Social Sciences, The University of Sydney, Camperdown, Sydney, New South Wales, Australia

* kcao3617@uni.sydney.edu.au

## Abstract

Clearing out zombie firms is a critical challenge for both developed and developing countries. This article draws upon data from Chinese listed SOEs to examine the impact of mixed-ownership reform on zombie firms. The findings indicate that non-state-owned shareholders participating in mixed-ownership reform by appointing directors can help reduce the possibility of SOEs becoming zombie firms, while participating in mixed-ownership reform through shareholding is not significant. Moreover, the impact of mixed-ownership reform on zombie firms is more pronounced for firms in competitive industries and firms located in the eastern region of China. Mechanism analysis reveals that the reduction of inefficient investment has a positive mediating effect between mixed-ownership reform and zombie firms.

## 1. Introduction

State-owned enterprises (SOEs) play an instrumental role in propelling China's economy toward high-quality development and structural transformation. However, a troubling trend has emerged where certain SOEs have transitioned into zombie firms—operating in a state of shutdown or semi-shutdown, enduring persistent losses, and relying solely on external assistance to survive [1]. Zombie firms not only occupy limited market resources and foster inefficient production capacities but also exert significant pressure on normal enterprises [2–4]. This strain inevitably leads to a decline in industry-wide productivity [5, 6], emerging as a debilitating "persistent disease" that significantly impedes the trajectory of high-quality economic development. Effectively clearing out zombie firms is critical for streamlining and improving the performance of SOEs, which needs more attention from government and academic institutions.

The study of zombie firms originated from an examination of Japan's "lost decade" in the 1990s. The earliest academic research that defined zombie firms was conducted by Hoshi (2006) [7] and Caballero et al. (2008) [5]. Zombie firms refer to businesses that are no longer profitable and can only survive with assistance from banks. In the case of Japan, with the deterioration of the economy, many Japanese companies were unable to even pay the interest on their debts. Under normal circumstances, these firms would have gone bankrupt. However, to

**Data Availability Statement:** All relevant data are within the manuscript and its Supporting Information files.

**Funding:** This research was funded by Zhejiang Federation of Humanities and Social Sciences

(2024N068) and Zhejiang Provincial Department of Education(Y202353848). The funder was not involved in the study design, data collection and analysis, decision to publish, or preparation of the manuscript.

avoid admitting to their shareholders that the loans were unlikely to be fully repaid, many banks allowed these struggling firms to suspend their payments. As a result, the banking industry kept these financially distressed firms afloat for an extended period of time. Zombie firms are not limited to Japan; they are prevalent in both developed and developing countries worldwide. Extensive research conducted by Adalet McGowan et al. (2018) has revealed a significant increase in the number of zombie firms across advanced economies following the Global Financial Crisis (GFC) [8]. Similarly, Banerjee and Hofmann (2022) [9] found a substantial rise in zombie firms, with their prevalence increasing from approximately 4% of all listed firms in the mid-1980s to as high as 15% in 2017 across 14 OECD countries. In Europe, the issue of zombie firms became particularly pronounced after the GFC and the subsequent European sovereign debt crisis. During this period, unconventional monetary policies led to weakly capitalized banks extending loans to struggling firms, a practice known as "evergreening" [10]. The introduction of Very Long-Term Refinancing Operations (VLTROs) further exacerbated the problem, as zombie firms obtaining new and larger loans from banks faced higher expected default probabilities [11]. In the context of the COVID-19 and slowdown in the global economy, central banks in various countries have implemented monetary easing policies, deepening the dependence of more enterprises on debt. The increasing number of zombie firms in the Eurozone and other countries has emerged as an urgent issue that requires immediate attention and resolution.

Presently, countries such as Japan, the United States, South Korea, and Germany recognize the social harm of blindly bankrupting zombie firms [12]. Instead, they emphasize the use of market mechanisms to revitalize idle resources, including land, capital, and technology. This approach involves redirecting underutilized resources from sectors with excess supply to those facing insufficient demand. It also entails shifting resources from inefficient sectors to more efficient ones, ultimately facilitating the resolution of zombie firms. By employing these strategies, economies can promote the efficient allocation of resources and encourage the gradual elimination of zombie firms. The Chinese government mainly adopts the following measures to eliminate zombie firms: government intervention and market mechanism. On one hand, Chinese governments have employed administrative measures to eliminate zombie firms. Provinces such as Guangdong, Zhejiang, Shanxi, Sichuan, and Henan implemented several policies aimed at eliminating zombie firms, yielding some positive results. However, local governments face a dual challenge of maintaining stable economic growth while mitigating risks. Zombie firms, despite their issues, contribute to stable tax revenues and employment, leading to complexities in initiating and executing their governance, as well as challenges in resettling affected personnel. On the other hand, market-oriented approaches have been advocated for clearing out zombie firms. In 2017, the Central Political Bureau meeting stressed the importance of effectively disposing of zombie firms, emphasizing greater reliance on market mechanisms for natural selection. The "14th Five-Year Plan" in 2021 reiterated the need to establish sustainable mechanisms for resolving overcapacity through market-oriented and rule-of-law approaches. Overall, both domestic and international policy practices suggest that market-oriented approaches may be a more effective method of clearing out zombie firms and facilitating their revival.

Mixed-ownership reform stands as a pivotal advancement in restructuring SOEs. The 19th National Congress of the Communist Party of China explicitly advocated for 'deepening the reform of SOEs, fostering a mixed-ownership economy, and nurturing globally competitive firms'. Additionally, the 2021 Government Work Report stressed the imperative to "comprehensively execute the three-year action plan for SOEs reform, fortify, enhance, and expand state-owned capital and firms, and deepen mixed-ownership reform". Mixed-ownership reform integrates non-state-owned shareholders through market-oriented and legally-

compliant methods, fostering a diversified corporate governance structure that facilitates the synergy of different ownership types, promoting collective growth [13]. However, existing literature inadequately explores whether mixed-ownership reforms can effectively clear out zombie firms and the specific mechanisms underpinning this potential impact. Addressing these inquiries and exploring their implications is essential not only for enriching theoretical studies concerning mixed-ownership reform and zombie firms but also for offering a fresh perspective on clearing out zombie firms and optimizing the efficiency of SOEs.

Drawing upon data spanning from 2008 to 2019 from Chinese listed SOEs, this article gathered information concerning the top ten shareholders' profiles, their ownership percentages, and the appointments of directors, supervisors, and senior executives. From the perspectives of ownership structure and directorial appointing directors, this article delves into the impact of mixed ownership reform on zombie firms. The primary findings are as follows: (1) non-state-owned shareholders introduced by mixed-ownership reform significantly reduce the possibility of SOEs becoming zombie firms via directorial appointments. However, the impact through shareholding is not notably substantial. This conclusion remains robust after a series of robustness tests. (2) Heterogeneous analysis indicates that the policy impact of mixed-ownership reform on zombie firms is more pronounced within competitive industries and the eastern region. (3) The curtailing inefficient investments serves as a positive mediating factor in the relationship between mixed-ownership reform and zombie firms.

This article provides several contributions as follows: (1) Provides a fresh perspective on clearing out zombie firms. Previous research primarily focused on external policies influencing the clearing out of zombie firms. However, the conclusions of this study indicates that non-state-owned shareholders appointing directors produce positive significance for clearing out zombie firms, offering a novel approach to clear out zombie firms. (2) Expands the scope of research on the policy effects of mixed-ownership reform. Relevant literature centered on the essence, motives, and impacts of mixed-ownership reform on micro-enterprises' innovation and investment, with limited exploration of its effects on zombie firms. This article empirically examines the impact of mixed-ownership reform from the perspectives of the ownership structure and board governance, enriching the research on its policy effects. (3) The findings of this study hold significant policy implications. Clearing out zombie firms should emphasize a "market-oriented primary approach supplemented by government intervention." Continuously introducing non-state-owned shareholders through market-oriented approaches to participate in the corporate governance of SOEs to enhance the governance and operational efficiency, ultimately leading to the clearance of zombie firms.

## 2. Literature review and hypothesis development

**2.1 Literature review.**    Relevant literature has explored the identification of zombie firms from the perspective of the CHK criterion [5], FN-CHK criterion [6], and the improved FN-CHK criterion [14]. Subsequently, some scholars have defined zombie firms from the perspectives of Firm age and value of interest coverage ratio [15], profitability and stock market valuation [9]. Additionally, scholars have discussed the causes of zombie firms from the angles of banks' motivations to conceal bad loans [7, 16–18] and government's public objectives (employment, taxation, social stability, etc.) [19–21]. Furthermore, some researchers have also delved into the dangers posed by zombie firms from the perspectives of employment [16], distortions in factor allocation [22–24], and increasing systemic financial risks [1].

Regarding the clearing out of zombie firms, developed nations such as Japan and the United States have employed strategies guided by the government and driven by market forces. They have initiated mergers, reorganizations, and bankruptcy procedures to rejuvenate businesses,

and re-establish employment opportunities, resulting in relatively successful results [25]. Some scholars in China advocate for the categorized management and resolution of zombie firms [26]: for firms displaying potential for turning losses into profits, governance strategies focus on strengthening management, undertaking technological transformations and upgrades, and implementing mergers and restructurings to revitalize the business. However, for firms that have completely or nearly lost profitability, the recommended approach involves bankruptcy and exit strategies. It can be inferred that the mechanisms of marketization and legalization offer hope for the revival of zombie firms into normal or even high-quality firms.

Research on mixed-ownership reform encompasses two primary areas of focus. Firstly, it delves into the essence [27], rationale [28] and the macroeconomic influence of mixed-ownership reform [29]. Secondly, it concentrates on its effects on micro-level enterprise aspects such as corporate innovation [30], corporate investment [31], and corporate performance [32]. Notably, studies demonstrate that mixed-ownership reform has diversified SOEs' equity, fostered equilibrium between state-owned and non-state-owned equity, alleviated two principal agency problems faced by SOEs, contributed to tightening SOEs' budget constraints, lessened policy-related burdens, and restructured incentive and career advancement mechanisms for SOEs executives [33].

While there is a sufficient literature on zombie firms and mixed ownership reform, there exist some gaps in the existing research. This article fills research gaps in two ways: Firstly, effectively addressing the issue of zombie firms is a critical challenge for both developed and developing countries worldwide. This article presents research findings indicating that the introduction of non-state-owned shareholders through mixed ownership reform has a positive impact on resolving zombie firms. Compared to adopting "external" economic policies to clear out zombie firms, this article provides evidence for clearing out zombie enterprises from the perspective of "within the enterprise" and supplements the literature on clearing out zombie firms. Secondly, unlike previous studies on developed economies, this article focuses on Chinese listed SOEs and empirically examines the impact of mixed ownership reform in transitional economies on zombie firms from the perspectives of equity structure and board governance. This has a profound impact on promoting sustainable economic growth.

## 2.2 Hypothesis development

Zombie firms exemplify the inefficiencies prevalent within SOEs. The origins of this inefficiency are commonly attributed to two main perspectives. One viewpoint suggests that the lack of internal property rights clarity and ownership vacancies are primary causes [34]. To address this, it advocates for property rights reform within SOEs, aiming to enhance internal corporate governance and ultimately improve SOEs' performance. Another viewpoint argues that extensive policy burdens imposed on SOEs lead to diverse operational objectives and soft budget constraints [35]. It advocates for divesting policy burdens from SOEs and establishing a fair competitive market environment to enhance SOEs' performance. Mixed-ownership reform stands as an intermediary approach between these viewpoints. On one hand, it facilitates equity ownership diversification, thereby refining corporate governance within SOEs. On the other hand, it aids in alleviating policy burdens, bolstering the status of market entities. Overall, mixed-ownership reform generates internal and external effects on zombie firms.

**2.2.1 The external effects of mixed-ownership reform on zombie firms.** Firstly, alleviate government interference. Excessive government intervention burdens State-Owned Enterprises (SOEs) with policy constraints, soft budget limitations, and inadequate market competitiveness, potentially leading previously thriving enterprises to become zombie firms [36, 37]. The introduction of non-state-owned capital significantly raises the government's intervention

costs in SOEs, thereby reducing unwarranted government interference [38]. This reduction aids in alleviating the policy burdens of SOEs, subsequently enhancing their performance. Consider labor costs as an illustration. The infusion of non-state-owned capital aids SOEs in optimizing labor resource allocation, adhering to profit-maximizing principles, and addressing the issue of 'surplus employees'. Despite potential continued government intervention post mixed-ownership reform, the augmented presence of non-state-owned capital and the rise in government intervention costs will gradually diminish undue governmental involvement.

Secondly, disrupt the alliance among the government, banks, and enterprises. The prolonged existence of zombie firms is advantageous for all three entities within the 'government, bank, and enterprise' nexus. Government subsidies and implicit assurances have eased the fiscal restrictions on SOEs, resulting in unwarranted credit expansion. As these zombie firms persist and expand, their exit costs rise, further entangling local governments and banks, fortifying the stability of this alliance. Mixed-ownership reform serves as a catalyst for a sequence of institutional reforms-such as the implementation of a professional manager system, management and employee shareholding, and an enhanced supervision and assessment framework-thereby curbing government subsidies and implicit assurances, thereby toughening the fiscal restrictions on zombie firms.

Thirdly, advance the marketization of factor distribution. Zombie firms not only monopolize market resources like land, labor, and credit, but also disrupt the market-oriented flow of these factors, constraining the survival opportunities for regular enterprises [39, 40]. Mixed-ownership reform promotes the reform of governance mechanisms through the form of "capital mixing", fosters a more market-driven flow of resources.

**2.2.2 The internal effects of mixed-ownership reform on zombie firms.** Firstly, SOEs exhibit problematic features such as the 'absence of owners' [13], leading to an excessive concentration of state-owned equity, inefficient corporate governance, and the potential for becoming zombie firms. Through mixed-ownership reform, the infusion of non-state-owned capital, inherently more market-oriented, diversifies internal equity structures within enterprises. This reform facilitates a balanced integration between state-owned and non-state-owned equity, effectively mitigating the prevailing principal-agent problems in SOEs [41].

Secondly, government intervention significantly impacts various aspects of SOEs, including personnel appointments, investment decisions, and incentive structures, hindering effective board governance. Mixed-ownership reform positively influences board governance in three key ways: (1) Increase the diversity of board members in SOEs. Presently, the number of independent directors within SOEs remains relatively low, often nominated by state-owned shareholders, occasionally leading to the selection of social celebrity figures as independent directors. This diminishes the professionalism and objectivity of independent director opinions [42]. Involvement of non-state-owned capital ensures the appointment of professional individuals as independent directors, thereby strengthening their role in corporate governance. (2) It helps to break the dual role situation in SOEs. Simultaneous holding of multiple positions fosters self-supervision, potentially leading to collusion between SOE executives and the board, resulting in decisions that undermine the interests of small and medium-sized shareholders [43]. Through mixed-ownership reform, to safeguard the interests of all stakeholders, breaking the existing dual-role situations will fortify the supervisory and governance functions of the board. (3) Reduce political connections. While political connections may offer benefits like easier financing and policy subsidies, they also attract more government intervention, adding policy burdens and reducing corporate governance efficiency [44, 45].

Thirdly, the incentive effect of SOEs executives. On one hand, executives within SOEs often straddle the roles of both "entrepreneurs "and "officials". Incentivization for these individuals extends beyond material rewards like salaries and equity incentives, with a significant

susceptibility to political advancement incentives [46]. The advancement of mixed-ownership reform holds the promise of revamping incentive structures for SOE executives. This initiative aims to bolster performance evaluations of executives linked to state-owned assets and gradually instate market-oriented incentive systems. On the other hand, executives within SOEs from non-state-owned backgrounds emerge through market-oriented selection processes, demonstrating a stronger inclination towards responding to salary incentives [47]. The surge in non-state-owned capital participation aids SOEs in fostering an effective corporate governance framework and markedly enhances governance efficiency. Based on the above analysis, this article proposes the following assumptions:

**H1:** Under certain other conditions, the participation of non-state-owned shareholders in mixed-ownership reform can help reduce the possibility of state-owned enterprises (SOEs) becoming zombie firms.

## 3. Model and data

### 3.1 Empirical methodology design

To verify the previous hypothesis and examine the impact of mixed-ownership reform on zombie firms, this article establishes the following regression model:

$$Zombie_{it} = \alpha_0 + \beta_1 Hungai_{it} + \beta_i Control_{it} + \sum Industry_j + \sum Year_t + \varepsilon_{it} \tag{1}$$

Where $Zombie_{it}$ measures whether it is a zombie firm of firm 'i' in year 't', $Hungai_{it}$ is quantified by the extent of mixed-ownership reform. $Control_{it}$ is control variables of firm 'i' in year 't', and $\varepsilon_{it}$ is the random disturbance term.

### 3.2 Variable definition

**3.2.1 Dependent variable.** Zombie firms ($Zombie_{it}$). Fukuda and Nakamura (2011) [6] further refined the CHK model by integrating profit criteria and perpetual loan standards, outlining the conditions for categorizing companies as zombie firms: (a) Pre-tax net profits falling below the market's lowest interest expenses; (b) Asset-liability ratios exceeding 50%; (c) Yearly loan amounts surpassing those of the preceding year. Despite the FN-CHK method's improvements in identifying zombie firms, potential inaccuracies remain. Total profits might include non-recurring gains or losses. If government subsidies are substantial, a company's profits could exceed the total minimum net interest expenses, leading to misidentification as a normal company. To enhance the precision of identifying zombie firms, this study, based on the FN-CHK approach, designates firms as zombie firms for the current year if their actual profits, after subtracting government subsidies, remain negative for three consecutive years.

**3.2.2 Independent variable.** Mixed-ownership reform ($Hungai_{it}$), involves not only the mixture at the equity level but also at the corporate governance level. This article measures the degree of non-state-owned shareholder participation in the mixed-ownership reform from two aspects: equity structure and board structure: (a) With regard to equity structure. According to Yang and Yin (2018) [48], this article systematically gathers and organizes data from various sources such as annual reports of listed SOEs, JvChao Net, CSMAR database to unveil information regarding the top ten shareholders. Each shareholder's nature is meticulously categorized into five distinct types: state-owned shareholders, foreign shareholders, private shareholders, institutional investors, and individual shareholders. The combined proportion of holdings attributed to foreign shareholders and private shareholders within the top ten shareholder ranks (Mix) serves as an indicator to gauge the degree of mixed-ownership reform. (b) With regard to board structure. The extent of mixed-ownership reform is gauged by the

percentage of directors appointed by non-state-owned shareholders within SOEs [49]. If a natural person shareholder serves as a director in a State-Owned Enterprise (SOE), they are considered a director appointed by the natural person shareholder. For corporate shareholders, the criterion is based on whether the director holds a position within the respective legal entity. This article adopts the ratio, denoted as 'Nonsoe_d,' representing the number of directors appointed by non-state-owned shareholders divided by the total number of directors, as a measure for evaluating mixed-ownership reform.

**3.2.3 Control variables.** According to Bai et al. (2009) [50], Fang and Sun (2019) [51] this article selects the following variables as control variables: company size (Size), leverage ratio (Lev), company age (Age), number of employees (Employee), and marketization degree (Market). Moreover, industry and yearly effects were factored in as control measures in the analysis.

## 3.3 Data source

The study focuses on analyzing the impact of mixed-ownership reform on zombie firms using data from SOEs listed on the Shanghai and Shenzhen stock markets from 2008 to 2019. The reasons for selecting Chinese listed SOEs as the research sample are as follows: (1) China, as one of the world's largest economies, is currently undergoing a critical period of economic transformation, marked by the implementation of policies such as mixed ownership reform. Studying the impact of mixed ownership reform on zombie firms holds significant importance for both developing and developed countries to eliminate zombie firms, and foster sustainable economic growth. (2) Non-listed SOEs are typically fully controlled by the government or state-owned enterprise groups, resulting in a relatively weaker level of mixed-ownership reform compared to listed SOEs. (3) Listed SOEs provide more recent and comprehensive data with timely information disclosure, audited by third-party institutions, thereby enhancing the reliability of research findings. (4) China's equity split reform was largely completed by the end of 2007, allowing non-state-owned capital to enter listed SOEs afterward. Therefore, the study commences the sample data from 2008 to capture this reform's impact.

According to Brandt et al. (2012) [40], Cai et al. (2018) [49] and Feng and Guo (2021) [52], the data is processed as follows: (a) Functional categories of SOEs are deleted, including military, power, telecommunications, petroleum (related to national economic security); water supply, heat, gas, and public transportation (related to natural monopoly industries). (b) Delete listed enterprises in financial industries such as securities, banking, and insurance; (c) Delete samples with abnormal numerical values, such as samples with an asset liability ratio greater than 1; (d) Delete the listed enterprises of *ST and ST, as these enterprises may have serious corporate governance issues.

This study identifies listed SOEs based on the nature of their actual controlling shareholders and measures the extent of mixed-ownership reform by quantifying the participation level of non-state-owned shareholders at both the equity and board of directors' levels. Due to the unavailability of relevant data directly from databases, the research relies on disclosed information in annual reports of listed companies. Specifically, it scrutinizes the top ten shareholders' details to discern the nature of each shareholder, their shareholding proportions, and the appointment of directors, eventually calculating the proportions of non-state-owned shareholders and directors appointed by them. Additionally, other pertinent variables were sourced from reputable databases such as CSMAR, CNRDS, and CCER Economic and Financial Database. Ultimately, the study collected observational data from 792 companies, and all data processing and calculations were conducted using Stata 17 software.

**Table 1. Descriptive statistics.**

| Variable | Obs | Mean | Std. Dev. | Min | Max |
|---|---|---|---|---|---|
| Zombie | 8134 | 0.184 | 0.387 | 0 | 1 |
| Mix | 8181 | 0.087 | 0.147 | 0 | 0.936 |
| Nonsoe_d | 7561 | 0.024 | 0.068 | 0 | 0.667 |
| Size | 9191 | 22.483 | 1.417 | 15.376 | 28.179 |
| Lev | 9191 | 0.528 | 0.203 | 0.002 | 1 |
| Age | 9191 | 2.684 | 0.425 | 0.693 | 3.401 |
| Employee | 9181 | 7.958 | 1.419 | 2.303 | 12.598 |
| Market | 9191 | 7.420 | 1.934 | 0.010 | 11.400 |

## 3.4 Summary description

The descriptive statistical data is displayed in **Table 1**. According to Table 1, the minimum and maximum values of Zombie are 0 and 1, while the standard deviation 0.387. This indicates that there exist significant differences among SOEs. At the same time, the mean value is 0.184, indicating a high degree of zombie firms in SOEs. The proportion of non-state-owned shareholders (Mix) in listed SOEs ranges from a minimum of 0 to a maximum of 0.936, with an average of 0.087. This indicates a disparity in the shareholding proportions of non-state-owned shareholders, generally leaning towards a lower proportion of ownership. Moreover, the proportion of directors appointed by non-state-owned shareholders (Nonsoe_d) ranges from a minimum of 0 to a maximum of 0.667, with an average of 0.024. This illustrates the involvement of non-state-owned shareholders in corporate governance remains relatively limited. The values of the remaining variables are within a reasonable range, and no outliers are observed.

# 4. Results and discussion

## 4.1 Baseline estimation results

**Table 2** reports the results of the impact of mixed-ownership reform on zombie firm. Column 1–2 report the impact of mixed ownership reform from the perspective of equity structure on zombie firms, while Column 3–4 are from the perspective of appointing directors.

Column 1 shows the regression coefficient of Mix is -0.026, and t value is -0.087, indicating that the coefficient of Mix is not significant. After adding control variables, the coefficient of Mix is 0.009, and t value is 0.33, also not significant. Furthermore, column 3 reports the regression coefficient of Nonsoe_d is -0.199 and the significance level is 1%. After adding control variables, the coefficient is -0.145 and the significance level is 1%. The above results indicate that holding shares of non-state-owned shareholders in SOEs is difficult to clear out zombie firms. Conversely, the appointment of directors by non-state-owned shareholders holds significant positive implications for clearing out zombie firms. That is to say, mixed-ownership reform contributes to the clearing out of zombie firms, and the impact is notably more substantial when non-state-owned shareholders appoint directors. Hence, this supports Hypothesis 1.

## 4.2 Robustness results

To enhance the credibility of the empirical findings, this article conducts robustness tests as follows:

**Table 2. Baseline estimation results.**

|  | (1) | (2) | (3) | (4) |
|---|---|---|---|---|
|  | zombie | zombie | zombie | zombie |
| Mix | -0.026 | 0.009 |  |  |
|  | (-0.887) | (0.330) |  |  |
| Nonsoe_d |  |  | -0.199*** | -0.145** |
|  |  |  | (-2.948) | (-2.217) |
| C | 0.406*** | 1.881*** | 0.381*** | 1.505*** |
|  | (9.056) | (17.263) | (8.548) | (13.521) |
| Controls |  | Yes |  | Yes |
| Year/Ind | Yes | Yes | Yes | Yes |
| Obs | 7,428 | 7,421 | 6,856 | 6,858 |
| Adj-$R^2$ | 0.067 | 0.176 | 0.078 | 0.152 |

Notes: t statistics in parentheses

* $p < 0.1$

** $p < 0.05$

*** $p < 0.01$

Zombie is the dependent variable. If a firm meets the FN-CHK standard and its actual profit minus government subsidies remains negative for three consecutive years, it is designated as a zombie company for that year, with zombie = 1; otherwise, zombie = 0. Mix is used to measure Mixed-ownership reform from the perspective of equity structure. Mix is the combined proportion of holdings attributed to foreign shareholders and private shareholders within the top ten shareholder ranks. Nonsoe_d is also used to measure Mixed-ownership reform form the perspective of corporate governance. Nonsoe_d is the ratio of the number of directors appointed by non-state-owned shareholders divided by the total number of directors. Year/Ind represents year and industry fixed effects. C is a constant term. Controls contains the following variables: Size is the logarithm of (1+total assets) (in RMB Yuan), Lev is the ratio of total debts to total assets, Age is the logarithm of (1+listed years), employee is the logarithm of (1+ number of employees), Market is marketization score of the city where the enterprise is located, and a higher score indicates a higher degree of marketization.

**4.2.1 Replace independent variable.** This article substitutes explanatory variables from both the equity structure and board structure perspectives. Firstly, it uses the proportion of shares held by the largest non-state-owned shareholders (Shr_nsoe1th) and the overall proportion of shares held by non-state-owned shareholders (Shr_nsoe) to replace Mix, measuring the mixed-ownership reform at the equity level. Secondly, it employs whether there are non-state-owned appointed directors in SOEs (If_nsoed) and the ratio of directors, supervisors, and senior managements appointed by non-state-owned shareholders (Nonsoe_djg) to replace Nonsoe_d, measuring the degree of mixed-ownership reform in corporate governance level. The regression results are presented in Table 3. The coefficients of Shr_nsoe1th and Shr_nsoe are not significant, while the coefficients of If_nsoedire and Nonsoe_djg are -0.029 and -0.170, with significance levels of 5% and 10%, indicating that mixed-ownership reform exhibits a positive impact on the clearing out of zombie firms thereby reaffirming the previous conclusion's robustness.

**4.2.2 Replace dependent variable.** This study initially redefines zombie firms by utilizing the criterion of the sum of actual profits, deducted from various subsidies for two consecutive years, resulting in a negative value (termed "zombie1"). The outcomes are presented column 1 of Table 4. Furthermore, as the policy effects from the involvement of non-state-owned shareholders in appointing directors may have a delayed effect, the dependent variables are each delayed by 1 period (termed "zombie2"). The regression results are displayed in column 2 of Table 4. The findings indicate that directors appointed by non-state-owned shareholders continues to demonstrate a substantial governance impact on zombie firms, affirming the robustness of the conclusions drawn in this study.

**Table 3. Robustness test: Replace independent variable.**

| Variables | (1) zombie | (2) zombie | (3) zombie | (4) zombie |
|---|---|---|---|---|
| Shr_nsoe1th | 0.025 | | | |
| | (0.315) | | | |
| Shr_nsoe | | -0.005 | | |
| | | (-0.087) | | |
| If_nsoed | | | -0.029** | |
| | | | (-2.361) | |
| Nonsoe_djg | | | | -0.170* |
| | | | | (-1.865) |
| C | 1.402*** | 1.400*** | 1.506*** | 1.502*** |
| | (11.051) | (11.049) | (13.535) | (13.494) |
| Controls | Yes | Yes | Yes | Yes |
| Year/Ind | Yes | Yes | Yes | Yes |
| Obs | 5,069 | 5,069 | 6,852 | 6,852 |
| Adj-$R^2$ | 0.147 | 0.147 | 0.153 | 0.152 |

Notes: t statistics in parentheses

* $p < 0.1$

** $p < 0.05$

*** $p < 0.01$

Zombie is the independent variable. Controls are shown in Table 2. Shr_nsoe1th is the proportion of shares held by the largest non-state-owned shareholders. Shr_nsoe is the ratio of the overall proportion of shares held by non-state-owned shareholders. Shr_nsoe1th and Shr_nsoe are applied to measure Mixed-ownership reform from the perspective of equity structure. If_nsoed represents whether there are non-state-owned appointed directors in SOEs, if so, If_nsoed == 1, otherwise If_nsoed == 0. Nonsoe_djg is the ratio of directors, supervisors, and senior managements appointed by non-state-owned shareholders.

**Table 4. Robustness test: Replace dependent variable.**

| | (1) zombie1 | (2) zombie2 |
|---|---|---|
| Nonsoe_d | -0.146*** | -0.079* |
| | (-2.114) | (-1.881) |
| C | 1.554*** | 1.372*** |
| | (9.658) | (11.690) |
| Controls | Yes | Yes |
| Year/Ind | Yes | Yes |
| Obs | 7,415 | 6,201 |
| Adj-$R^2$ | 0.112 | 0.131 |

Notes: t statistics in parentheses

* $p < 0.1$

** $p < 0.05$

*** $p < 0.01$

Zombie1 and Zombie2 is the independent variable. If a firm meets the FN-CHK standard and its actual profit minus government subsidies remains negative for two consecutive years, it is designated as a zombie company for that year, with zombie1 = 1; otherwise, zombie1 = 0. Zombie2 represents a lag period of Zombie. Controls are shown in Table 2.

**Table 5. Robustness test: IV-2sls.**

|  | (1) | (2) |
|---|---|---|
|  | Nonsoe_d | Zombie |
| Nonsoe_d |  | -0.375* |
|  |  | (-1.94) |
| ANonsoe_d | 0.992*** |  |
|  | (19.310) |  |
| Controls | Yes | Yes |
| Year/Ind | Yes | Yes |
| C-DW | 863.576 |  |
| Sargan P | 0.227 |  |
| Obs | 7555 | 6852 |
| Adj-$R^2$ | 0.136 | 0.099 |

Notes: t statistics in parentheses

* $p < 0.1$

** $p < 0.05$

*** $p < 0.01$

Column1-2 show the regression results of IV-2sls model. ANonsoe_d is instrumental variable, measured by the average proportion of directors appointed by non-state-owned shareholders in other listed SOEs within the same industry and year. Controls are shown in Table 2.

**4.2.3 Instrumental variable regression (IV-2sls).** SOEs are significantly influenced by government intervention, and improvements in their operational conditions may not solely result from the impact of mixed-ownership reform but could also be affected by macroeconomic policies, economic cycles, and other factors. To eliminate endogeneity and potential biases in the research findings, this study utilizes the average proportion of directors appointed by non-state-owned shareholders in other listed SOEs within the same industry and year (ANonsoe_d) as an instrumental variable to reexamine the impact of mixed-ownership reform on zombie firms. On one hand, an increasing of directors appointed by non-state-owned shareholders in other listed companies within the same industry signifies a more substantial involvement of these shareholders in the mixed-ownership reform. This suggests potentially improved corporate governance conditions within SOEs in that industry. On the other hand, there is no apparent correlation between director appointments in other enterprises and the resurgence of zombie firms. The results are displayed in **Table 5**.

The column1 of **Table 5** shows that the P-value of the Sargan statistic for the instrumental variable ANonsoe_d is 0.227, greater than 10%, respectively, accepting the exogenous hypothesis of the instrumental variable. The weak instrumental variable test (C-DW value is 83.576, greater than 10) indicates that the instrumental variable has passed the weak instrumental variable test. The column2 of **Table 5** shows that the coefficient of Nonsoe_d is significantly negative, indicating that the previous conclusion is robust after alleviating endogeneity issues.

### 4.3 Heterogeneity analysis

**4.3.1 Industry heterogeneity.** The impact of mixed-ownership reform on zombie firms might differ significantly. Monopolistic industries, typically found in sectors like steel and coal, pose a formidable challenge for mixed-ownership reform: Firstly, monopolistic industries wield substantial control over production factors and corporate governance, resulting in increased information asymmetry and hidden barriers to entry for non-state-owned. This

**Table 6. Heterogeneity analysis: Industry and regional heterogeneity.**

|  | (1) | (2) | (3) | (4) |
|---|---|---|---|---|
|  | **Monopoly** | **Competitive** | **East** | **Middle-west** |
| Nonsoe_d | -0.170 | -0.146** | -0.141* | -0.064 |
|  | (-1.312) | (-1.978) | (-1.889) | (-0.499) |
| C | 1.482*** | 1.509*** | 1.602*** | 1.687*** |
|  | (7.439) | (12.227) | (9.110) | (8.957) |
| Controls | Yes | Yes | Yes | Yes |
| Year/Ind | Yes | Yes | Yes | Yes |
| Obs | 2,782 | 4,076 | 4,042 | 2,816 |
| Adj-R$^2$ | 0.139 | 0.166 | 0.136 | 0.200 |

Notes: t statistics in parentheses

* $p < 0.1$

** $p < 0.05$

*** $p < 0.01$

Monopoly and competitive industries are distinguished by the level of competition in the industry. East and middle-west are divided by the spatial location of the city where the enterprise is located. Zombie is the dependent variable. Nonsoe_d is used to measure Mixed-ownership reform form the perspective of corporate governance. Nonsoe_d is the ratio of the number of directors appointed by non-state-owned shareholders divided by the total number of directors. Controls are shown in Table 2.

creates difficulties in effectively counterbalancing the influence of state-owned shareholders. Secondly, executives in monopolistic SOEs often exhibit traits akin to "quasi-government officials," strengthening their incentives for "political promotion." This inclination might prompt them to shoulder more policy burdens, potentially overlooking the operational needs of non-state-owned shareholders. Based on this, sample firms are divided into monopolistic and competitive groups to explore the varied effects of mixed-ownership reform on zombie firms. Regression conclusions are detailed in **Table 6**.

The results of columns 1 and 2 in **Table 6** indicate that the coefficients of mixed-ownership reform in both monopolistic and competitive industries are -0.17 and -0.146, while the coefficient in monopolistic industries is not significant. It suggests that the involvement of non-state-owned shareholders in corporate governance plays a more pronounced role in addressing zombie firms within competitive industries. This might be attributed to the inherent nature of monopolistic industries, which hold greater control over production factors and corporate governance, along with the stronger resemblance of top-level management in state-owned enterprises to "quasi-government officials." These factors might limit the efficacy of mixed-ownership reforms. Therefore, crafting and refining mixed-ownership reform policies should be more specific and tailored according to the characteristics of different industry types.

**4.3.2 Regional heterogeneity.** The economic disparities among regions in China can impact the effectiveness of mixed-ownership reforms on addressing zombie firms. This study divides sampled companies based on their operating provinces into eastern, central, and western regions. Regression findings in columns 3 and 4 of **Table 6** reveal that the coefficient of mixed-ownership reform is negative in both the eastern and central-western regions, yet statistically insignificant in the latter. This suggests a more pronounced impact of mixed-ownership reforms on zombie firms in the eastern region.

Research by Nie et al. (2016) [14] notes that economically developed eastern regions tend to have a lower proportion of zombie firms compared to less-developed central-western regions. Concerning the reform proportions, approximately 54.78% of companies in the eastern region, while figures were 57.32% in the central and 51.61% in the western regions. Despite the lower

percentage in the western region, the disparity is relatively minor. One plausible explanation is the weak economic footing in these areas, characterized by single-industry structures, which complicates governing zombie firms due to increased challenges and uncertainties. Economic underdevelopment in the central-western regions complicates dealing with zombie firms, involving the legitimate rights of various stakeholders, potentially disrupting industry stability and triggering financial risks. Consequently, local governments might exhibit hesitation in clearing out zombie firms for SOEs play a more significant role in regional stability and development.

## 4.4 Mechanism analysis

Local governments often burden state-owned enterprises (SOEs) within their jurisdiction with policy objectives like employment and social stability, subjecting them to soft budget constraints, and leading to generally lower investment efficiency compared to private enterprises [53]. When facing losses, SOEs commonly receive subsidies from local governments to support their survival. Studies indicate that when profitable, SOEs tend to engage in excessive investment, a tendency more pronounced in local SOEs than in central enterprises [54]. Owing to biased industrial policies, government subsidies, and local protectionism, SOEs tend to expand indiscriminately and overinvest, deviating from market principles. These actions often trigger overcapacity issues and foster the emergence of zombie firms [55].

The introduction of non-state-owned capital through mixed-ownership reform has been shown to have a significantly positive impact on investment efficiency, profitability, and innovation capabilities [56, 57]. Hence, does reducing inefficient investments have a positive moderating role between mixed-ownership reform and zombie firms? This paper establishes the following model:

$$M_{it} = \beta_0 + \beta_1 Hungai_{it} + \beta_i Control_{it} + \sum Industry_j + \sum Year_t + \vartheta_{it} \qquad (2)$$

$$Zombie_{it} = \sigma_0 + \sigma_1 Hungai_{it} + \sigma_2 M_{it} + \sigma_i Control_{it} + \sum Industry_j + \sum Year_t + \pi_{it} \qquad (3)$$

$M_{it}$ represents the mediating variable, inefficient investment (Non_invest). In addition, this article refers to Richardson (2006) [58] to calculate inefficient investment. The specific calculation formula is as follows:

$$Invest_{it} = \alpha_0 + \beta_1 Invest_{it-1} + \beta_2 Size_{it-1} + \beta_3 EBIT_{it-1} + \beta_4 Grow_{it-1} + \beta_5 Age_{it} + \mu_i$$
$$+ \sum Industry_j + \sum Year_t + \varepsilon_{it} \qquad (4)$$

$Invest_{it}$ represents the new added investment of firm 'i' in year 't', expressed as (net fixed assets of this year—net fixed assets of the previous year)/total assets; $Size_{it-1}$ represents the total assets in year 't-1'; $EBIT_{it-1}$ represents the net profit before interest and tax for year 't-1'; $Grow_{it-1}$ represents the growth rate of operating revenue in year 't-1'; $Age_{it}$ represents the age of the enterprise.

The residual derived from eq 5 signifies the inefficient investment level. A positive residual value indicates an excessive investment, surpassing the optimal level for the enterprise. Conversely, a negative residual value signifies insufficient investment. Adhering to Zhu et al. (2020) [59], this study utilizes the absolute value of the residual (Non_investment) to depict the enterprise's degree of inefficient investment. A higher absolute value reflects a greater magnitude of inefficiency in the enterprise's investment practices. The regression results are shown in Table 7.

Table 7. Mechanism analysis: Inefficient investment.

| | (1) | (2) | (3) |
|---|---|---|---|
| | zombie | Non_invest | zombie |
| Nonsoe_d | -0.145** | -0.016* | -0.129* |
| | (-2.217) | (1.919) | (-1.926) |
| Non_invest | | | 0.375*** |
| | | | (7.439) |
| C | 1.505*** | -0.183*** | 1.402*** |
| | (13.521) | (-7.483) | (13.279) |
| Controls | Yes | Yes | Yes |
| Year/Ind | Yes | Yes | Yes |
| Obs | 6,858 | 7,333 | 6,852 |
| Adj-R² | 0.152 | 0.089 | 0.160 |

Notes: t statistics in parentheses

* p < 0.1

** p < 0.05

*** p < 0.01

Column 1–3 show the regression results of mediating Effect Model. Zombie is the dependent variable. Controls are shown in Table 2. Nonsoe_d is the independent variable. Non_invest is the moderating variable, which is measured by the absolute value of the residual of regression model from Richardson (2006).

The coefficient of Nonsoe_d in column 2 are significantly negative. In column 3, the coefficient of Nonsoe_d is significantly negative, while the coefficient of Non_invest is significantly positive. This indicates a significant mediating effect of Non_invest between mixed-ownership reform and zombie firms.

## 5. Discussion

The results obtained in the previous article are mainly discussed in this section. Firstly, the results show that the coefficient of mixed-ownership reform to zombie firms is significantly negative, indicating that mixed-ownership reform can help reduce the possibility of state-owned enterprises (SOEs) becoming zombie firms. In reality, the issue of addressing zombie firms is of paramount importance for both developed and developing economies. However, related literature remains relatively limited. Existing studies have explored the ways to address zombie firms through several lenses, including relationship banking [60], bank competition [61], FinTech [62], digital transformation [63] and The Belt and Road Initiative [64]. Moreover, the research conducted by Ernesto et al. (2022) [65] offers valuable insights into the effectiveness of institutional reforms in Portugal, an OECD country grappling with a high prevalence of zombie firms. Contrary to conventional wisdom, the findings suggest that not all zombie firms are inherently doomed and the right institutional framework in place can facilitate their restructuring and eventual recovery, which could be more beneficial than simply forcing them into bankruptcy. Furthermore, Fischer (2021) [60] and Carreira et al. (2022) [66] support these findings by highlighting the positive impact of restructuring strategies in preventing the emergence of zombie firms and facilitating their revitalization. Given this, will the heterogeneous shareholder equity integration and resource integration brought about by mixed-ownership reform help clear out zombie firms? As a developing country in transition, China's SOEs play a crucial role in economic development, but at the same time, the risk of evolving into zombie firms is also higher [67]. Based on this, this article takes Chinese listed SOEs as the object and confirms the positive impact of mixed ownership reform on reducing

the risk of zombification in SOEs, thus offering valuable theoretical insights into clear out zombie firms.

Secondly, research by Leire et al. (2022) [68] and Veganzones and Severin (2023) [69] has demonstrated that enhancing corporate governance can mitigate the prevalence of zombie firms. However, this article extends "corporate governance" to " corporate governance of mixed-ownership enterprises". Focusing on Chinese listed State-Owned Enterprises (SOEs), the study evaluates mixed-ownership reform by assessing the proportion of non-state share-holders appointing directors (supervisors and executives) relative to all directors. The findings indicate that non-state shareholders' direct involvement in appointing directors to SOEs sig-nificantly diminishes the zombification risk of SOEs, whereas mere participation in sharehold-ing does not yield the same effect. The appointment of directors by non-state-owned shareholders significantly enhances the diversity of SOEs' boards, reduces the prevalence of dual roles, decreases government entanglement, and effectively bolsters governance efficiency. Moreover, it alleviates policy burdens on SOEs and diminishes the likelihood of them becom-ing zombie firms.

Thirdly, the findings suggest that the reduction of inefficient investment plays a crucial mediating role in the relationship between mixed-ownership reform and zombie firms. Previ-ous studies have highlighted the propensity of Chinese SOEs to experience heightened govern-ment intervention, coupled with subsidies, biased industrial policies, and local government protection. These factors often contribute to excessive investment and diminished investment efficiency [53]. This article introduces inefficient investment as a mediator variable to delve deeper into the relationship of "mixed-ownership reform, inefficient investment, and zombie firms", thereby enriching existing theoretical frameworks. The observed results may stem from the pronounced profit orientation of non-state-owned shareholders, who actively engage in the governance of SOEs, thereby enhancing investment efficiency and reducing the likelihood of SOEs transitioning into zombie firms. In addition, different from prior research conducted on large and medium-sized enterprises in Germany [60] and small and medium-sized enter-prises in Portugal [66], this article chooses SOEs (mainly medium-sized and large enterprises) as a sample and has value for existing literature. Moreover, the findings also underscore the heightened impact of mixed-ownership reform within competitive industries and the eastern regions of China. This discovery holds substantial policy implications, highlighting the absence of a one-size-fits-all solution for addressing zombie firms. Instead, a tailored approach akin to targeted drugs is imperative for effectively tackling this issue.

## 6. Conclusions

The Chinese economy is undergoing a pivotal phase of structural transformation, wherein the presence of zombie firms poses a significant obstacle to its health and long-term sustainability. Examining the effect of mixed ownership reform on zombie firms holds paramount impor-tance in identifying strategies to eradicate them and foster a path towards enduring economic development. Based on data from Chinese listed SOEs, this article examines the effect of mixed-ownership reform on zombie firms, yielding the following conclusions: Firstly, non-state-owned shareholders participating in mixed ownership reform by appointing directors can help reduce the possibility of SOEs becoming zombie firms, while participating in mixed ownership reform through shareholding is not significant. This conclusion remained robust after a series of rigorous tests. Secondly, mixed-ownership reform exhibits a more significant inhibitory effect on zombie firms in competitive industries and the eastern region of China. Thirdly, inefficient investment plays an intermediary role between mixed-ownership reform

and zombie firms. This means that mixed ownership reform reduces the likelihood of zombie firms by reducing inefficient investment, which is beneficial for clearing out zombie firms.

Based on the conclusions of this study, several policy recommendations are discussed as follows. Firstly, emphasizing a market-driven approach is crucial for effectively promoting mixed-ownership reform. Mere amalgamation of state-owned and private ownership is insufficient; it is imperative to actively engage non-state-owned shareholders in the governance of State-Owned Enterprises (SOEs), empowering them with greater influence to fully realize the policy objectives of mixed-ownership reform. Secondly, clearing out zombie companies is a global challenge: providing assistance to firms on the verge of bankruptcy can inadvertently perpetuate their zombie status. Conversely, withholding assistance may deny potentially viable businesses the chance to recover. Furthermore, research indicates that even recovered zombie firms tend to exhibit weak resilience and are susceptible to relapse into zombie status [9]. The conclusion of this article indicates that market-oriented and legal measures should be adopted to promote the integration of state-owned and non-state-owned capital, improve corporate governance capabilities and competitiveness, and thus achieve the clearance of zombie firms. Thirdly, cultivate a fair and competitive market environment, and treat state-owned and private capital fairly in market access, administrative approval, licensing, and bidding. At the same time, actively promote the mixed-owned reform in monopolistic industries and the central and western regions, fully unleash positive effects on clearing zombie firms. Fourthly, the conclusion of this article indicates that improving the investment efficiency of state-owned enterprises helps to prevent and resolve zombie firms. For state-owned enterprises, it is necessary to actively integrate the advantages of state-owned and non-state-owned capital, establish a more specialized investment project management team, improve investment efficiency, and reduce the possibility of becoming a zombie firm.

Although this article provides a comprehensive examination of the relationship between mixed ownership reform and zombie firms, there are still some limitations and directions for further exploration. Firstly, it's important to acknowledge that this article's focus on state-owned enterprises as research samples may introduce bias due to the exclusion of non-listed state-owned enterprises. This limitation could influence regression results and potentially skew conclusions. Additionally, the absence of consideration for the impact of COVID-19 on zombie firms suggests a potential subjectivity in the research findings. Therefore, we encourage future studies to broaden their scope by incorporating data from both developing and developed economies over longer temporal and spatial spans to provide a more comprehensive analysis. Secondly, while the mixed ownership reform entails the integration of state-owned and non-state-owned capital, future research could explore the effects of state-owned capital's involvement in the corporate governance of private enterprises, particularly in addressing issues related to private zombie firms. Lastly, bankruptcy represents a significant mechanism for clearing out zombie firms. Further research could delve into strategies for mitigating the employment and social stability challenges that accompany the bankruptcy of zombie firms.

## Supporting information

**S1 Data. The data is used in this article for empirical analysis.**
(ZIP)

## Acknowledgments

We would like to thank the editor and anonymous reviewers for their constructive comments and suggestions. All errors are our own.

## Author Contributions

**Conceptualization:** Yufei Yin.

**Data curation:** Yufei Yin.

**Formal analysis:** Yufei Yin.

**Investigation:** Kexin Cao.

**Methodology:** Kexin Cao.

**Resources:** Kexin Cao.

**Software:** Kexin Cao.

**Writing – original draft:** Yufei Yin.

**Writing – review & editing:** Kexin Cao.

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
