## [Decision Letter · Decision Letter 0]

15 Jan 2024

PONE-D-23-39861Mixed-Ownership Reform and the Governance of Zombie Firms: Evidence from ChinaPLOS ONE

Dear Dr. Cao,

Thank you for submitting your manuscript to PLOS ONE. After careful consideration, we feel that it has merit but does not fully meet PLOS ONE’s publication criteria as it currently stands. Therefore, we invite you to submit a revised version of the manuscript that addresses the points raised during the review process.

We look forward to receiving your revised manuscript.

Kind regards,

Ionela Munteanu, PhD

Academic Editor

PLOS ONE

Journal Requirements:

Zhejiang Federation of Humanities and Social Sciences （2024N068）

Zhejiang Provincial Department of Education （Y202353848）

3. We note that your Data Availability Statement is currently as follows: Data are available from all authors of this article

Additional Editor Comments :

You investigate the potential of mixed-ownership reform to promote the governance of zombie firms, invoking certain other conditions. Previous studies suggest that the prevalence of zombie firms is influenced by monetary policies, tax policies, interest rates, economic stability among others. Focused on the Chinese context, the current study presents evidence that mixed-ownership reform can influence zombie firms. You suggest a novel method to illustrate that the appointment of directors by non-state-owned shareholders can effectively combat zombie firms.

I found your research interesting since it offers a comprehensive analysis within the Chinese context regarding the appointment of shareholders and introduces a set of compelling approaches to achieve your research objectives. Given your research focus on China, you need to spend some efforts to explain (1) why the Chinese context is particularly relevant to your research question and (2) whether the results can be generalized to other empirical settings. You may want to consider the following studies that offer different approaches to the determinants or conditions of zombie firms: https://ssrn.com/abstract=3829038, https://doi.org/10.7441/joc.2023.02.05, https://doi.org/10.1016/j.japwor.2023.101188.

In addition, please consider improving the following: (3) In the caption of Table 2 and subsequent please explain the significance of numbers in paratheses; (4) insert reference to the limitations of the study before the conclusion section; (5) Please double check the English writing, there are some errors in the text.

To facilitate the publication review process, I strongly recommend that the authors also provide comprehensive responses to all the recommendations formulated by the two referees, listed below.

Reviewers' comments:

Reviewer's Responses to Questions

**Comments to the Author**

1. Is the manuscript technically sound, and do the data support the conclusions?

Reviewer #1: Yes

Reviewer #2: Yes

2. Has the statistical analysis been performed appropriately and rigorously? 

Reviewer #1: Yes

Reviewer #2: Yes

3. Have the authors made all data underlying the findings in their manuscript fully available?

Reviewer #1: Yes

Reviewer #2: Yes

4. Is the manuscript presented in an intelligible fashion and written in standard English?

Reviewer #1: Yes

Reviewer #2: Yes

5. Review Comments to the Author

Reviewer #1: Dear Authors,

It was a real pleasure to read and review your research paper. I appreciate the research idea and how did you construct the paper in order to reach the proposed research objectives. But, in my opinion I would reconsider the link between the proposed hypothesis and the model from eq. (1) (also where you defined the dependent variable). Your proposed hypothesis states that: Under certain other conditions, mixed-ownership reform can promote the governance of zombie firms. So, in this conditions the governance of zombies would be the dependent variable, and from the eq (1) the dependent variable is Zombie (a composite score variable - I guess) that describe a status based on Pre-tax net profits, Asset-liability ratios and Yearly loan amounts. Based on these, I don't find a link between this status and governance. Please try to answer this question: how do you measure governance of zombie firms and what is the link between those metrics and the Zombie variable. Also, please motivate the sample period selection: why 2019 and not the current year.

Kind regards,

The Anonymous Reviewer

Reviewer #2: The manuscript addresses a very interesting topic, but it can be published after making some adjustments. The main adjustments I recommend are:

1. Authors must use the impersonal style (to give up expressions like ...our study...we...etc.).

2. The authors must present, in the introduction, the existing situation worldwide and briefly present the European experience.

3. At the end of the literature review section, the authors must insist on identifying a research gap that justifies the completion of this study.

4. The authors must justify in the paper the choice of the country.

5. The authors must insert a discussions section. In the discussion section, the authors must interpret the results obtained in the context of similar studies that confirm or refute their results.

6. More additional references can be used in the introduction, LR, discussions sections of the paper (Blažková, I., & Chmelíková, G. (2022). Zombie Firms during and after Crisis. Journal of Risk and Financial Management, 15(7), 301.; Rodano, G., & Sette, E. (2019). Zombie firms in Italy: a critical assessment. Bank of Italy Occasional Paper, (483).; Banerjee, R., & Hofmann, B. (2022). Corporate zombies: Anatomy and life cycle. Economic Policy, 37(112), 757-803.; El Ghoul, S., Fu, Z., & Guedhami, O. (2021). Zombie firms: Prevalence, determinants, and corporate policies. Finance Research Letters, 41, 101876.)

7. The authors must improve the conclusions section taking in account the quality of the study and extend the policy implications section.

8. The authors must present the limits of the research and the future research directions

6. PLOS authors have the option to publish the peer review history of their article (what does this mean?). If published, this will include your full peer review and any attached files.

Reviewer #1: No

Reviewer #2: No

---

## [Author Response · Author response to Decision Letter 0]

18 Feb 2024

Additional Editor Comments:

1. Given your research focus on China, you need to spend some efforts to explain (1) why the Chinese context is particularly relevant to your research question.

Answer: Thank you for your suggestions. Choosing China as our research object is based on the following considerations: clearing out zombie companies is a challenge for both developed and developing countries. Unlike zombie firms in developed countries, Zombie firms in China have typical characteristics of government intervention. At the same time, China, as one of the world's largest economies, is currently undergoing a critical period of economic transformation, marked by the implementation of policies such as mixed ownership reform. Studying the impact of mixed ownership reform on zombie firms holds significant importance for both developing and developed countries. Different from previous literature on using external macroeconomic policy adjustments to clear out zombie firms, this article may provide a way to clear out zombie firms from the perspective of mixed-ownership reform within enterprises. 

2. whether the results can be generalized to other empirical settings. You may want to consider the following studies that offer different approaches to the determinants or conditions of zombie firms: https://ssrn.com/abstract=3829038, https://doi.org/10.7441/joc.2023.02.05, 

https://doi.org/10.1016/j.japwor.2023.101188.

Answer: Thank you for providing us with the references, which are very helpful to us. We believe the results in the article can be generalized to other empirical settings. The key point to clear out zombie firms lies in enhancing their competitiveness. There are two issues that need to be solved: one is who is a zombie firm, and the other is how to effectively clean up zombie firms. Firstly, this article proposes an improved identification standard of zombie firms based on the FN-CHK standard, which helps to improve the accuracy of identifying zombie firms. Secondly, this article confirms that the participation of non-state-owned shareholders in the corporate governance of state-owned enterprises can reduce government intervention, strengthen the competitiveness of enterprises, and thus help to clear out zombie firms. The conclusion of this article has significant implications for the clearing out of zombie firms in developed and developing countries. 

The firs literature (https://ssrn.com/abstract=3829038) builds the RDT model to examinate of the risks taken on by a company during the insolvency process. We believe that it makes a useful supplement to the conclusions of this article. Because not all zombie firms need to go bankrupt, some firms can be revived through some rescue measures (such as restructuring). The first literature helps to determine whether a firm is restructuring or going bankrupt. The conclusion of this article is helpful for how to restructure and leverage the advantages of heterogeneous shareholders to enhance the competitiveness. The second literature (https://doi.org/10.7441/joc.2023.02.05), soft constraint constraints are an important cause of zombie firms in Japan's manufacturing industry. The conclusions of this article show that mixed-ownership reform helps to breaking soft constraint constraints, which is beneficial for clearing out zombie firms. In the third literature (https://doi.org/10.1016/j.japwor.2023.101188), a firm is defined as a zombie firm with an interest coverage ratio of below one over the previous three years. The above literature has provided a foundation for identifying firms. However, it must be acknowledged that due to differences in financial system structure and financial environment, the identification standards for zombie firms still need to be continuously improved. This is a direction that needs to be studied on in the future.

3. In the caption of Table 2 and subsequent please explain the significance of numbers in paratheses;

Answer: Thank you for your suggestions. We have added the explanation for the significance of numbers in paratheses in the caption of Table 2 and subsequent. 

4. Insert reference to the limitations of the study before the conclusion section;

Answer: Thank you for your suggestions. We have added the limitations of the study and the future research directions as follows:

Although this article provides a comprehensive examination of the relationship between mixed ownership reform and zombie firms, there are still some limitations and directions for further exploration. Firstly, it's important to acknowledge that this article's focus on state-owned enterprises as research samples may introduce bias due to the exclusion of non-listed state-owned enterprises. This limitation could influence regression results and potentially skew conclusions. Additionally, the absence of consideration for the impact of COVID-19 on zombie firms suggests a potential subjectivity in the research findings. Therefore, we encourage future studies to broaden their scope by incorporating data from both developing and developed economies over longer temporal and spatial spans to provide a more comprehensive analysis. Secondly, while the mixed ownership reform entails the integration of state-owned and non-state-owned capital, future research could explore the effects of state-owned capital's involvement in the corporate governance of private enterprises, particularly in addressing issues related to private zombie firms. Lastly, bankruptcy represents a significant mechanism for clearing out zombie firms. Further research could delve into strategies for mitigating the employment and social stability challenges that accompany the bankruptcy of zombie firms.

5. Please double check the English writing, there are some errors in the text.

Answer: Thank you for your suggestions. We have carefully checked and corrected the errors and inappropriate expressions in the paper. In future research, we will pay special attention to this point. Thank you again.

Reviewer #1

1. In my opinion, I would reconsider the link between the proposed hypothesis and the model from eq. (1) (also where you defined the dependent variable). Your proposed hypothesis states that: Under certain other conditions, mixed-ownership reform can promote the governance of zombie firms. So, in these conditions the governance of zombies would be the dependent variable, and from the eq (1) the dependent variable is Zombie (a composite score variable - I guess) that describe a status based on Pre-tax net profits, Asset-liability ratios and Yearly loan amounts. Based on these, I don't find a link between this status and governance. Please try to answer this question: how do you measure governance of zombie firms and what is the link between those metrics and the Zombie variable.

Answer: Thank you for your suggestions very much! We have reconsidered the meaning of the term "governance" and found that it does not accurately match the content of this article. The variable of "zombie firm" in this article is a 0-1 variable. The empirical regression results of this article show that the regression coefficient of mixed-ownership reform on zombie firms is significantly negative, and mixed-ownership reform helps to improve the efficiency of state-owned enterprises, thereby reducing the possibility of SOEs becoming zombie firms. In practical scenarios, a cohort of zombie firms such as Shandong Yanmei International Coking and Youyi Special Steel have embraced a union of "state-owned framework and private mechanisms" through mixed-ownership reform, igniting the resource integration capabilities of SOEs and invigorating market mechanisms within private enterprises. For instance, Shandong Yanmei International Coking achieved a notable milestone in the year of mixed-ownership reform (2020) by realizing a total profit of 502 million yuan, signifying a staggering 12% year-on-year growth, while concurrently reducing its debt-to-asset ratio from 104% in 2017 to 61.54% in 2020. Therefore, this article uses the term "governance" to show the positive effect of mixed-ownership reform on zombie firms. However, based on your suggestion, we believe that a more accurate expression of the hypothesis in this article should be that " Under certain other conditions, mixed-ownership reform helps to reduce the possibility of state-owned enterprises (SOEs) becoming zombie firms". At the same time, we have also made corresponding modifications in the article. Thank you again!

2. Please motivate the sample period selection: why 2019 and not the current year.

Answer: Thank you for your question! The selection of sample period is mainly based on the following two considerations: (1) After 2019 means COVID-19. It causes enormous harm to the global economy, and the Chinese economy is no exception. The slowdown in macroeconomic growth may give rise to more zombie firms. After the COVID-19, it is still necessary to wait for enough time to test whether zombie firms that have carried out mixed-ownership reform before the COVID-19 or during the COVID-19 have stronger recovery capacity. At present, the impact of the COVID-19 epidemic on zombie firms is rare in the existing literature, and it is challenging to accurately assess the resilience of zombie firms brought about by the mixed-ownership reform, which is also a complex problem that is worth continuing to study. We also state this in the limitations of this article. (2) This article measures the degree of non-state-owned shareholder participation in the mixed-ownership reform from equity structure and board structure. This requires manual collection and identification of information on the top ten shareholders from existing databases, including key data such as shareholding ratios and shareholder nature. Due to difficulties in data, our team has only collected data before 2019, and we are working hard on data after 2019. Thank you again!

Reviewer #2: 

1. Authors must use the impersonal style (to give up expressions like ...our study...we...etc.).

Answer: Thank you for your suggestions. We have changed the expressions like …our study…we…etc into the impersonal style. In future writing, we will also pay special attention to this point. Thank you!

2. The authors must present, in the introduction, the existing situation worldwide and briefly present the European experience.

Answer: Thank you for your suggestions. We have added the existing situation worldwide and the European experience in the second paragraph of the introduction section. This section is necessary as it contributes to the coherence of the article. In future writing, we will pay attention to this point. The modified content is as follows: 

The study of zombie firms originated from an examination of Japan's "lost decade" in the 1990s. The earliest academic research that defined zombie firms was conducted by Hoshi (2006) and Caballero et al. (2008). Zombie firms refer to businesses that are no longer profitable and can only survive with assistance from banks. In the case of Japan, with the deterioration of the economy, many Japanese companies were unable to even pay the interest on their debts. Under normal circumstances, these firms would have gone bankrupt. However, to avoid admitting to their shareholders that the loans were unlikely to be fully repaid, many banks allowed these struggling firms to suspend their payments. As a result, the banking industry kept these financially distressed firms afloat for an extended period of time. Zombie firms are not limited to Japan; they are prevalent in both developed and developing countries worldwide. Extensive research conducted by Adalet McGowan et al. (2018) has revealed a significant increase in the number of zombie firms across advanced economies following the Global Financial Crisis (GFC). Similarly, Banerjee and Hofmann (2022) found a substantial rise in zombie firms, with their prevalence increasing from approximately 4% of all listed firms in the mid-1980s to as high as 15% in 2017 across 14 OECD countries. In Europe, the issue of zombie firms became particularly pronounced after the GFC and the subsequent European sovereign debt crisis. During this period, unconventional monetary policies led to weakly capitalized banks extending loans to struggling firms, a practice known as "evergreening" (Acharya et al., 2019). The introduction of Very Long-Term Refinancing Operations (VLTROs) further exacerbated the problem, as zombie firms obtaining new and larger loans from banks faced higher expected default probabilities (Christian et al., 2021). In the context of the COVID-19 and slowdown in the global economy, central banks in various countries have implemented monetary easing policies, deepening the dependence of more enterprises on debt. The increasing number of zombie firms in the Eurozone and other countries has emerged as an urgent issue that requires immediate attention and resolution.

3. At the end of the literature review section, the authors must insist on identifying a research gap that justifies the completion of this study.

Answer: Thank you for your suggestions. We have added the research gap in the literature review section as follows: 

While there is a sufficient literature on zombie firms and mixed ownership reform, there exist some gaps in the existing research. This article fills research gaps in two ways: (1) Effectively addressing the issue of zombie firms is a critical challenge for both developed and developing countries worldwide. This article presents research findings indicating that the introduction of non-state-owned shareholders through mixed ownership reform has a positive impact on resolving zombie firms. Compared to adopting "external" economic policies to clear out zombie firms, this article provides evidence for clearing out zombie enterprises from the perspective of "within the enterprise" and supplements the literature on clearing out zombie firms. (2) Unlike previous studies on developed economies, this article focuses on Chinese listed SOEs and empirically examines the impact of mixed ownership reform in transitional economies on zombie firms from the perspectives of equity structure and board governance. This has a profound impact on promoting sustainable and healthy economic development.

4. The authors must justify in the paper the choice of the country.

Answer: Thank you for your suggestions. We have added the discussion about the choice of the country in “3.3 Date source”. Your suggestions have helped us improve the comprehensiveness of the article. Thank you again! The revised content is as follows:

The reasons for selecting Chinese listed SOEs as the research sample are as follows: (1) China, as one of the world's largest economies, is currently undergoing a critical period of economic transformation, marked by the implementation of policies such as mixed ownership reform. Studying the impact of mixed ownership reform on zombie firms holds significant importance for both developing and developed countries to eliminate zombie firms, and foster sustainable and healthy economic development. 

5. The authors must insert a discussions section. In the discussion section, the authors must interpret the results obtained in the context of similar studies that confirm or refute their results.

Answer: Thank you for your suggestions. We have inserted a discussion in the paper. The discussion section is very useful for making the paper more comprehensive. The modified content is as follows:

The results obtained in the previous article are mainly discussed in this section. Firstly, the results show that the coefficient of mixed-ownership reform to zombie firms is significantly negative, indicating that mixed-ownership reform can help reduce the possibility of state-owned enterprises (SOEs) becoming zombie firms. In reality, the issue of addressing zombie firms is of paramount importance for both developed and developing economies. However, related literature remains relatively limited. Existing studies have explored the ways to address zombie firms through several lenses, including relationship banking[60], bank competition[61], FinTech[62], digital transformation [63] and The Belt and Road Initiative[64]. Moreover, the research conducted by Ernesto et al. (2022) [65]offers valuable insigh

---

## [Decision Letter · Decision Letter 1]

11 Mar 2024

The impact of mixed-ownership reform on zombie firms: evidence from Chinese listed SOEs

PONE-D-23-39861R1

Dear Dr. Cao,

We’re pleased to inform you that your manuscript has been judged scientifically suitable for publication and will be formally accepted for publication once it meets all outstanding technical requirements.

Kind regards,

Ionela Munteanu, PhD

Academic Editor

PLOS ONE

Reviewers' comments:

Reviewer's Responses to Questions

**Comments to the Author**

1. If the authors have adequately addressed your comments raised in a previous round of review and you feel that this manuscript is now acceptable for publication, you may indicate that here to bypass the “Comments to the Author” section, enter your conflict of interest statement in the “Confidential to Editor” section, and submit your "Accept" recommendation.

Reviewer #1: All comments have been addressed

Reviewer #2: All comments have been addressed

2. Is the manuscript technically sound, and do the data support the conclusions?

Reviewer #1: Yes

Reviewer #2: Yes

3. Has the statistical analysis been performed appropriately and rigorously? 

Reviewer #1: Yes

Reviewer #2: Yes

4. Have the authors made all data underlying the findings in their manuscript fully available?

Reviewer #1: Yes

Reviewer #2: Yes

5. Is the manuscript presented in an intelligible fashion and written in standard English?

Reviewer #1: Yes

Reviewer #2: Yes

6. Review Comments to the Author

Reviewer #1: It was a pleasure to read and review your paper and I encourage you to continue the research in the field.

Also, I agree with your responds and I consider that this paper could be published in this form.

Reviewer #2: The authors took into account the recommendations made and considerably improved the manuscript. Considering the adjustments made by the authors, I recommend the publication of the manuscript.

7. PLOS authors have the option to publish the peer review history of their article (what does this mean?). If published, this will include your full peer review and any attached files.

Reviewer #1: No

Reviewer #2: No

---

## [Editor Report · Acceptance letter]

24 Jun 2024

PONE-D-23-39861R1 

PLOS ONE

Dear Dr. Cao, 

I'm pleased to inform you that your manuscript has been deemed suitable for publication in PLOS ONE. Congratulations! Your manuscript is now being handed over to our production team.

Kind regards, 

on behalf of

Dr. Ionela Munteanu 

Academic Editor

PLOS ONE